# Seasonal Trends in Suicide Attempts-Keywords Related Searches: A Google Trends Analysis

**DOI:** 10.3390/healthcare12131273

**Published:** 2024-06-26

**Authors:** Krzysztof Bartosz Klimiuk, Dawid Krefta, Michał Krawczyk, Łukasz Balwicki

**Affiliations:** 1Department of Public Health and Social Medicine, Faculty of Health Sciences, Medical University of Gdańsk, 80-210 Gdańsk, Poland; balwicki@gumed.edu.pl; 2Faculty of Electronics, Telecommunications and Informatics, Gdansk University of Technology, 80-233 Gdańsk, Poland; 3Faculty of Medicine, Medical University of Gdańsk, 80-210 Gdańsk, Poland

**Keywords:** suicide, Google Trends, seasonal patterns

## Abstract

Suicide is a significant public health concern globally, with its varying rates influenced by numerous factors, including seasonal changes. Online search behaviors, particularly searches related to suicide and mental health, have been proposed as real-time indicators of suicidal ideation in populations. In this study, a cross-sectional time series analysis was conducted, utilizing data on suicide attempts from the Polish Police Headquarters and online search behavior from Google Trends over a decade. Suicide attempt data were analyzed alongside the frequency of Google searches for suicide-related keywords derived from the Polish Corpus of Suicide Notes. A total of 66 keywords were selected for analysis to identify seasonal trends and patterns in search behavior. The study employed linear regression, Seasonal Mann-Kendall tests, and TBATS models to analyze the data. Suicide rates show seasonal patterns, peaking in warmer months. However, keyword searches did not strongly correlate with peak suicide months. This study enhances our understanding of suicide-related search trends and their potential connection to suicide rates. It suggests avenues for more effective prevention efforts and the potential for future algorithms to predict suicide rates and identify at-risk groups.

## 1. Introduction

The term “suicide” encompasses several definitions, with the simplest being the deliberate act of taking one’s own life [1,2]. According to the World Health Organization (WHO) data, approximately 700,000 people commit suicide annually, making it the fourth leading cause of death among individuals aged 15–29 [3].

Strategies for suicide prevention may encompass various approaches, including environmental measures, psychotherapeutic interventions, pharmacological interventions, and multi-level strategies [4]. Meta-analyses have demonstrated that multidimensional medical interventions can effectively mitigate the risk of suicide [4]. As suicide rates in the United States have risen nearly 30% from 1999 to 2016 [5] and pose a significant threat to individuals aged 15–29 [3], it is crucial to approach the topic from a prevention perspective. Detecting warning signs, such as often talking or writing about death, dying, or suicide, and making comments about feeling hopeless, helpless, or worthless, is important [6]. The analysis of the language and vocabulary of individuals attempting suicide allows for understanding the thought process of the suicidal person and, consequently, offers a chance to anticipate such an attempt [7].

Google Trends is a free tool developed by Google to identify the most frequently searched words and phrases on its search engine platform [8]. Google Search is among the most widely used search engines, particularly in Poland [9]. Google Trends finds extensive applications in various industries, including marketing [10]. It provides insights into frequently searched terms, the popularity of phrases over time, user interests, regional variations in phrase popularity, and trends in phrase popularity [10]. In recent years, the medical field has shown increasing interest in this tool [11]. It enables us to identify the phrases individuals use when searching for symptoms of their illnesses or potential treatments [11].

Google Trends has been used in many different infodemiology studies for analysis [8,12,13,14,15]. It has also revealed correlations between the frequency of certain search phrases and mental health conditions and addiction rates [12]. Researchers have made attempts to correlate specific search phrases with fluctuations in suicide rates [12,16]. It is worth noting that suicide rates exhibit annual seasonality [17], and multiple algorithms have been employed in different countries to predict spikes in suicide rates [16,18,19].

Although Google Trends currently lacks robust validation [20] for predicting behavioral disorders, there is potential for further research to enhance content analysis and the prediction of human behavior.

Prior studies have demonstrated the utility of using Google Trends as a tool to predict suicide rates [16], emphasizing the value of sparse word counts for assessing suicide prevention effectiveness. Our study uses Google Trends to investigate the relationship between suicide rates and online search behavior based on keywords from suicide notes, focusing on Poland. The study aims to provide a novel approach to understanding seasonal patterns in suicide attempts using Google Trends.

## 2. Materials and Methods

### 2.1. Materials

The authors of this study conducted a cross-sectional time series analysis to discern seasonal patterns among keywords. This analysis relied on the Google Trends tool, which allows us to track the relative frequency of searches over time in a specific location, quantified as the relative search volume (RSV). An RSV of 0 indicates minimal or no searches for the term during the specified period, while an RSV of 100 signifies the term’s peak popularity. For instance, if a term was searched with a frequency equal to 50% of the maximum searches, its RSV would be 50.

We obtained data on the number of suicide attempts during specific months and quarters from the Polish Police Headquarters in July 2021. The data collection methods evolved over time: prior to 2012, data was gathered quarterly, while from 2013 onwards, it was collected on a monthly basis. The authors sourced keywords for our study from the Polish Corpus of Suicide Notes [21], resulting in a total of 857 words. With support from the National Prosecutor’s Office in 2008, many suicide notes from Poland were gathered and organized for statistical analysis [7].

### 2.2. Data Collection

Data collection took place in December 2021, with each keyword entered individually into Google Trends. Each word from the database of suicide notes was individually entered into Google Trends (https://trends.google.com/trends/, accessed on 10 December 2021.) with set parameters (time: January 2010 to December 2020, location: Poland). The data were downloaded in CSV format for analysis. Each keyword was searched and retrieved individually. All collected data were meticulously compiled and organized in an Excel spreadsheet. From the initial pool of 857 time series, authors excluded 791, ultimately selecting 66 keywords for our analysis. A detailed diagram of the word selection process is presented in Figure 1.

### 2.3. Statistical Analysis

We utilized various statistical techniques to analyze the seasonal patterns of suicide attempt-related search terms from Google Trends. The data was not transformed, and R 4.0.5 was used for statistical calculations [22]. [NO_PRINTED_FORM] Linear regression was used to estimate the slope, which represents the change in RSV per year. Seasonal Mann-Kendal tests were performed to investigate significant secular trends in time series data. Classical Seasonal Decomposition by Moving Averages was used to extract the seasonal components. TBATS models from the forecast package were used to determine significant seasonal periods [23], which are designed to forecast time series with multiple seasonal periods.

## 3. Results

According to the data obtained from the Polish Police Headquarters, the month with the highest number of suicide attempts is June, while the month with the lowest number of attempts is February. Similarly, the second quarter of the year (April–June) has the highest number of suicide attempts, while the first quarter (January–March) has the lowest. These findings suggest a seasonal component to the occurrence of suicide attempts, with a higher incidence during the warmer months of the year. These findings regarding the number of suicide attempts by month and quarter have been presented in Table 1.

The data in Table 2 were sorted by month with the highest seasonal component [RSV], and additionally, the keywords with the lowest seasonal component [−10.0 < RSV] were highlighted and bolded. All keywords have a TBATS seasonal period of 12, indicating yearly seasonality. Quarterly data on suicides were not used to calculate months with the highest and lowest seasonal component [RSV]. In Table 2, the Seasonal Mann-Kendall trend tau is used to identify and measure monotonic trends in the time series data while accounting for seasonal variations, with positive values indicating increasing trends and negative values indicating decreasing trends.

Our study involved an analysis of the most frequently searched suicide attempt-related terms over the past decade using Google (as shown in Table 2). Through this analysis, we were able to identify seasonal patterns in the search volumes for various verbs, adjectives, and nouns. The results of our analysis are presented in Table 2, providing a summary of the time series for each term. The data in Table 2 have been sorted by month with the highest seasonal component [RSV], and additionally, the keywords with the lowest seasonal component [−10.0 < RSV] have been highlighted and bolded. All keywords have a TBATS seasonal period of 12, indicating yearly seasonality. Quarterly data on suicides were not used to calculate months with the highest and lowest seasonal component [RSV].

## 4. Discussion

To the best of our knowledge, this research pioneers the use of Google Trends to establish correlations between search trends and suicide attempts, utilizing keywords derived from suicide notes. This investigation addresses a critical gap in understanding the relationship between the volume of suicide attempts-related Google searches and national suicide attempt rates. Another novelty of this study is that the database of analyzed words is significantly larger than those used in other studies leveraging Google Trends for suicide research [14,24] or infodemiology in general [13,15].

The months with the highest suicide attempt rates are June, May, and July, while the lowest rates occur in February, January, and November. Interestingly, none of the keywords that were searched seem to correlate significantly with these key months. For instance, the word “lear” is most frequently searched in April and least frequently in December, one month before the spike and decline in suicide attempts.

In February, when suicide attempts rates reach their lowest, there is a noticeable surge in searches for words with positive connotations, such as “love”, “safety”, “fantastic”, “pretty”, and “wonderful”. On the other hand, in January there is a cluster of words frequently appearing in a sexual context, including “lust”, “fidelity”, “temptation”, “moral”, alongside words describing physical appearance like “ideal”, “normal”, and “delicate”. Notably, June and May show no significant relationships with any specific search keywords.

Police data showed a reduction in suicide attempts during the February period when search terms such as “love”, “safety”, “pretty”, “wonderful”, and “fantastic” are prominent in Google searches. It is necessary to consider the impact of Valentine’s Day, which occurs in February, on searches for similar keywords [25]. This may indicate a favorable period for individuals in relationships and families, as these interpersonal bonds are well-recognized protective factors against suicide [26]. However, it is important to exercise caution in drawing such definitive conclusions solely based on the data presented. During May, which ranks as the second highest month for suicide attempts, searches related to work and stress increased, suggesting a potential link with unemployment and stress-related illnesses as known suicide risk factors [26].

Our research predominantly focused on identifying novel search terms. The authors conducted an extensive analysis of a large number of entries, assessing whether any of them had a direct connection with suicide attempts. In contrast to other cited studies [12,27], which revealed links to suicide attempts with keywords like “depression”, “divorce”, “unemployment”, and complex phrases like “suicide guide”, our study explored a broader spectrum of words. For example, a study from Taiwan [28] identified 37 relevant entries featuring the aforementioned terms. While authors examined a greater number of words, they did not delve into terms with previously established correlations. Furthermore, our study encompasses a more extensive array of terms compared to previous research conducted in Poland [27], employing the methodology grounded in Google Trends.

Identifying specific time frames when searches for suicide attempt-related words peak could significantly simplify the task of reaching individuals at risk [27]. A nuanced understanding of online behaviors has the potential to greatly enhance the effectiveness of prevention strategies [11].

In the context of public health, given the Internet’s role as a medium for health-related inquiries [29], one practical approach could involve search engine providers implementing suitable filters and detection mechanisms to identify potentially harmful sources in keyword-driven search results [28].

While data retrieved from Google Trends and similar search engine databases can never replace traditional data collection methodologies for population health, further refinement and continued constructive dialogue between researchers and technology companies like Google could transform search engine data into a potent, real-time resource. This has the potential to effectively monitor shifts in public health and health-related behaviors within our society [8].

There are inherent limitations to our investigation, which are common in research of this nature. Despite the notable correlation effects demonstrated in similar studies, their validity is generally considered low [20]. Furthermore, the analytical techniques employed in this paper do not allow for predictions at the individual level. Moreover, due to the fact that the keywords are based on suicide notes from Poland, it would be difficult to repeat the study in other countries based on similar resources.

The data employed in our study also come with certain limitations. Prior research indicates that the recorded data on suicide attempts in Poland may be underrepresented [30]. The sociodemographic details of those who made the searches were not accessible. Additionally, Google Trends does not provide exact quantitative data but rather relative percentage data, making it impossible to discern the actual frequency of a specific search phrase [8].

It is also worth noting that keywords cannot be used out of context, and the authors indicate that many words, despite the demonstrated statistical relationship with suicide attempts, do not seem to be related to them. An example would be the word “heavy” (“ciężki”) or “favorite” (“ulubiony”).

Lastly, it is crucial to emphasize that suicide attempts are a multifaceted phenomenon driven by the conjunction of numerous factors. Therefore, the present research has limited external validity as it examines specific factors [31]. This study, as a separate component of the overall analysis of internet user behavior, is more fundamental in nature rather than applied.

## 5. Conclusions

The findings from our research provide a better understanding of search trends associated with suicide and the potential connection between search terms and suicide attempt rates. The analysis of the gathered data revealed distinct seasonal shifts in search terms tied to suicide, related to periods of both high and low suicide attempts. This identification of specific words lays a foundation for coming up with conclusions that may boost efforts to prevent suicide.

Identifying an increased number of suicide-related online searches might allow for deploying targeted crisis intervention and mental health awareness campaigns more effectively. This conclusion is valid only if one assumes that search terms related to suicide attempts are identical to the keywords from suicide notes. In the future, by creating appropriate algorithms, it may be possible to predict the number of suicide attempts, prevent them, predict the potential number of suicide attempts, and identify specific groups at risk [27].

## Figures and Tables

**Figure 1 healthcare-12-01273-f001:**
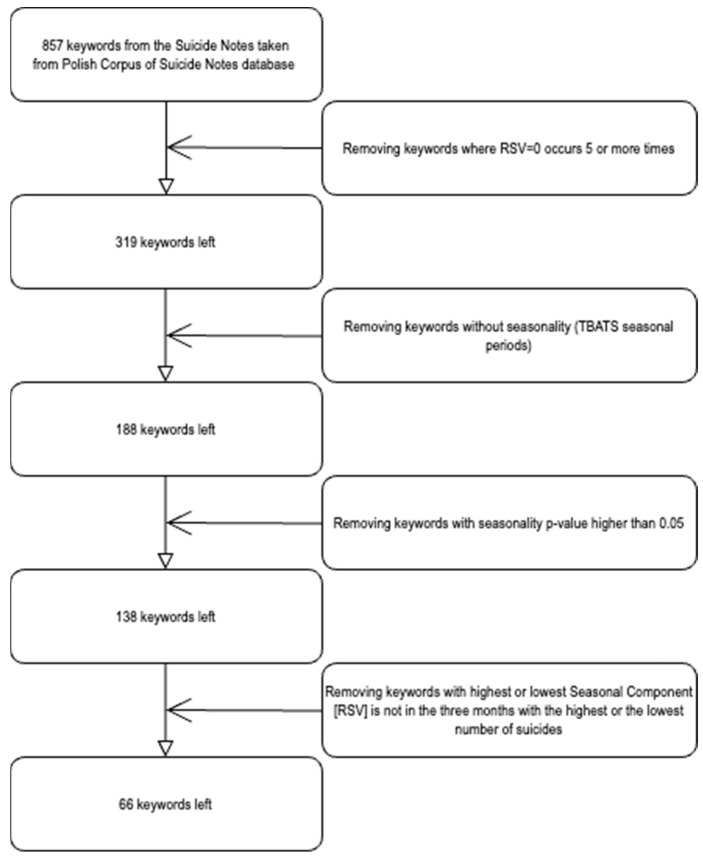
Flowchart illustrating the keyword selection process from the initial pool of 857 keywords to the final set of 66 keywords used in the analysis.

**Table 1 healthcare-12-01273-t001:** Number of suicide attempts by month and quarter. Data was obtained from the Polish Police Headquarters.

	January	February	March	April	May	June	July	August	September	October	November	December
2010	1288			1518			1293			1357		
2011	1060			1378			1380			1306		
2012	1359			1477			1348			1607		
2013	509	522	666	795	778	823	800	707	720	782	714	759
2014	748	715	826	910	937	867	958	899	874	867	785	821
2015	832	725	858	886	921	871	934	883	735	741	775	812
2016	757	759	864	877	865	915	842	784	795	813	768	822
2017	851	841	941	915	1043	970	999	983	835	911	887	963
2018	941	743	978	1060	989	1007	969	1041	816	896	848	879
2019	971	900	996	1000	1054	1118	1041	1064	901	993	958	965
2020	1002	966	883	862	1069	1163	1105	1081	995	988	884	1015
Sum—Months	6611	6171	7012	7305	7656	7734	7648	7442	6671	6991	6619	7036
Sum—Quarters	23,501			27,068			25,782			24,916		

**Table 2 healthcare-12-01273-t002:** Results of seasonal analysis of Google Trends search terms.

Names	Slope [RSV/Year] (lm)	Seasonal Mann Kendal Trend Tau	Month with the Highest Seasonal Component [RSV]	Month with the Lowest Seasonal Component [RSV]
**Correct (prawidłowy)**	**0.64**	**0.21**	**March [25.45]**	**August [−23.33]**
Mercy (miłosierdzie)	1.12	0.39	April [25.17]	August [−7.94]
Passion (pasja)	0.83	0.36	April [24.61]	July [−7.81]
**Common (pospolity)**	**1.49**	**0.44**	**May [24.17]**	**February [−14.40]**
Adoration (adoracja)	3.26	0.68	April [23.77]	August [−8.63]
Expensive (drogi)	0.57	0.29	March [22.38]	December [−6.87]
**Interest (zainteresowanie)**	**−1.64**	**−0.22**	**May [19.45]**	**July [−12.53]**
**Nervous (nerwowy)**	**−0.96**	**−0.21**	**May [17.42]**	**August [−23.39]**
**Safe (bezpieczny)**	**2.06**	**0.34**	**February [16.81]**	**December [−12.33]**
**Complex (złożony)**	**2.02**	**0.38**	**April [16.51]**	**August [−17.62]**
**Normal (normalny)**	**−2.16**	**−0.45**	**January [16.18]**	**August [−15.64]**
Active (aktywny)	2.55	0.58	March [15.88]	July [−9.24]
Ideal (idealny)	−0.64	−0.19	January [14.24]	June [−6.06]
**Rough (szorstki)**	**2.66**	**0.36**	**May [13.79]**	**November [−10.32]**
**Lust (żądza)**	**−2.2**	**−0.30**	**January [13.65]**	**August [−10.23]**
**Security (bezpieczeństwo)**	**−2.09**	**−0.51**	**May [13.23]**	**August [−24.22]**
**Favorite (ulubiony)**	**−1.01**	**−0.22**	**May [12.00]**	**July [−18.09]**
Strange (dziwny)	−0.9	−0.22	March [11.77]	August [−7.70]
**Boring (nudny)**	**1.47**	**0.21**	**May [11.67]**	**September [−12.12]**
**Survive (przeżyć)**	**−1.61**	**−0.34**	**March [11.63]**	**August [−21.24]**
Ready (gotowy)	1.18	0.26	May [11.49]	October [−8.91]
**Love (miłość)**	**−1.43**	**−0.28**	**February [11.40]**	**July [−28.53]**
Prejudice (uprzedzenie)	1.53	0.38	June [11.32]	October [−9.05]
**Subtle (subtelny)**	**0.68**	**0.16**	**May [11.28]**	**December [−12.42]**
**Aspirations (aspiracje)**	**−3.57**	**−0.46**	**May [11.13]**	**August [−14.18]**
**Impression (wrażenie)**	**−1.27**	**−0.20**	**April [10.84]**	**August [−17.02]**
**Moral (moralny)**	**−2.42**	**−0.36**	**January [10.62]**	**August [−13.22]**
Illusion (złudzenie)	−4.88	−0.65	May [10.35]	October [−7.67]
Regret (żal)	−3.38	−0.60	February [10.32]	July [−8.03]
**Worry (martwić)**	**3.38**	**0.40**	**April [10.28]**	**September [−17.57]**
**Recognition (uznanie)**	**1.53**	**0.33**	**June [9.79]**	**December [−12.60]**
Friendship (przyjaźń)	−1.94	−0.49	January [9.73]	August [−5.30]
Heavy (ciężki)	1.51	0.24	April [9.59]	July [−7.82]
Panic (panika)	0.61	0.17	March [9.57]	August [−3.50]
Unpleasant (nieprzyjemny)	3.09	0.37	January [9.29]	December [−7.19]
Dream (marzenie)	−3.31	−0.64	January [9.19]	June [−7.43]
**Ambition (ambicja)**	**−2.12**	**−0.28**	**January [8.73]**	**August [−10.20]**
**Tenderness (czułość)**	**−1.41**	**−0.21**	**June [8.64]**	**August [−11.04]**
Sad (smutny)	−0.83	−0.30	June [8.54]	August [−3.95]
Temptation (pokusa)	1.28	0.48	January [8.46]	March [−4.97]
Shock (wstrząs)	1.04	0.30	January [8.44]	July [−6.75]
Trust (zaufanie)	−2.83	−0.58	April [8.34]	August [−4.93]
Admiration (uwielbienie)	2.35	0.32	June [8.33]	December [−9.97]
Need (potrzeba)	0.9	0.17	April [8.29]	July [−9.47]
Call (zew)	−1.68	−0.58	February [8.12]	August [−4.76]
Natural (naturalny)	3.65	0.83	May [7.98]	July [−9.81]
**Nerves (nerwy)**	**1.34**	**0.28**	**March [7.72]**	**July [−14.49]**
Curious (ciekawy)	−1.64	−0.42	January [7.61]	August [−4.95]
**Overt (jawny)**	**1**	**0.15**	**January [7.60]**	**August [−10.71]**
Tragic (tragiczny)	−1.03	−0.60	May [7.37]	February [−3.01]
Hatred (nienawiść)	−2.3	−0.50	January [7.27]	July [−4.59]
Fear (trwoga)	−1.69	−0.17	March [7.13]	July [−9.00]
Joy (radość)	−1.13	−0.24	April [7.07]	July [−9.38]
Loneliness (samotność)	−5.54	−0.80	April [7.04]	July [−9.67]
Wonderful (wspaniały)	−0.97	−0.28	February [6.52]	August [−6.86]
Pretty (ładny)	−1.62	−0.28	February [6.46]	September [−7.37]
Aware (świadomy)	−2.82	−0.60	January [6.23]	April [−6.38]
Wild (dziki)	2.29	0.64	January [5.91]	March [−3.76]
Colorless (bezbarwny)	4.61	0.82	April [5.43]	January [−6.95]
Fantasy (fantazja)	−2.08	−0.41	February [5.20]	December [−4.95]
Fidelity (wierność)	−2.01	−0.46	January [5.04]	August [−5.16]
Delicate (delikatny)	4.93	0.67	January [4.86]	May [−5.22]
Clear (jasny)	4.67	0.78	April [4.40]	December [−7.82]
Negative (negatywny)	2.06	0.45	March [4.38]	July [−6.16]
Dangerous (niebezpieczny)	3.67	0.66	April [3.83]	September [−5.07]
Fantastic (fantastyczny)	−1.64	−0.38	February [2.25]	July [−3.27]

## Data Availability

The original contributions presented in the study are included in the article/Appendix A, further inquiries can be directed to the corresponding author/s.

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
