# Peer review of "Seasonal Trends in Suicide Attempts-Keywords Related Searches: A Google Trends Analysis"

_healthcare, 2024, doi:10.3390/healthcare12131273_

Round 1

Reviewer 1 Report

Comments and Suggestions for Authors

The paper is interesting, but needs improvements.

(1) line 19 of page 1, TBATS is used once, it is better to give the full name.

(2) the contribution is not clearly claimed in introduction.

(3) Maybe it is a template problem, the paper does not contain the line label on the right.

(4) Tables 1 and 2 appear to be drawn outside the margins and not centered.

(5) paper title should not end with “.”. "Keywords" should be followed by ":".

(6) the caption of figure 1 is missed.

(7) Some search terms in the table do not have data items, need to explain the reason or delete.

some additional comments:

The paper lacks scientific nature. For experiment design, the authors do not say the pros and cons of their design against others. The topic has been studied by other researchers. The authors do not show their strengthen against previous studies. What is the difference or novelty of this paper compared with existing papers?

Comments on the Quality of English Language

The English language is ok.

Author Response

Dear Reviewer,

Thank you for your detailed feedback on our manuscript. We have carefully considered your suggestions and made the following improvements:

Line 19 of page 1, TBATS is used once, it is better to give the full name.
We have clarified what an acronym TBATS is at the end of section 2.3, "Statistical analysis". We believe it is better not to elaborate on such details in the abstract.

The contribution is not clearly claimed in the introduction.
We have added a sentence in the introduction to highlight the contribution of this study. Additionally, the authors' contributions are detailed at the end.

Maybe it is a template problem, the paper does not contain the line label on the right.
The template has been used.

Tables 1 and 2 appear to be drawn outside the margins and not centered.
Tables 1 and 2 have been adjusted to fit within the margins as per the template guidelines.

Paper title should not end with “.”. "Keywords" should be followed by ":".
The period has been removed, and a colon has been added after "Keywords".

The caption of figure 1 is missed.
The caption for Figure 1 has been added.

Some search terms in the table do not have data items, need to explain the reason or delete.

All data in Table 2 have been completed. The issue might have been caused by the layout of the terms in two languages (English and Polish).

Additional comments:

The paper lacks scientific nature. For experiment design, the authors do not say the pros and cons of their design against others. The topic has been studied by other researchers. The authors do not show their strength against previous studies. What is the difference or novelty of this paper compared with existing papers?

A sentence has been added to the first paragraph of the discussion to further clarify the novelty of this study compared to previous research.
We appreciate your constructive feedback and believe these changes have strengthened our manuscript.

Best regards,

Krzysztof Bartosz Klimiuk

Reviewer 2 Report

Comments and Suggestions for Authors

This original paper introduces the use of Google Trends to establish correlations between search trends and suicide attempts, utilizing keywords derived from suicide notes. 

Attached, I have noted spelling/grammar corrections that need to be made to Figure 1. Also, the Figure legend is missing.

I appreciate that the authors clearly state the limitations of this study, particularly since the content of the suicide notes could not be linked to actual individual searches, rather than general trends provided by the corresponding content. 

In general, it is well written with sound methodology and the conclusions follow from the study.

Author Response

Dear Reviewer,

Thank you very much for your kind words and thorough review of our manuscript. We have corrected the spelling and grammatical issues in Figure 1 and added an appropriate caption. Specifically, the following changes were made:

  1. "form" was corrected to "from".
  2. "keywors" was corrected to "keywords".
  3. "whit" was corrected to "with".
  4. "RSV is in the is" was corrected to "RSV is not in the".

We appreciate your feedback and are glad that you found the methodology sound and the conclusions well-supported by the study.

Best regards,

Krzysztof Bartosz Klimiuk

Reviewer 3 Report

Comments and Suggestions for Authors

1. "Strategies for suicide prevention may encompass various approaches, including en-32 vironmental measures, psychotherapeutic interventions, pharmacological interventions, 33 and multi-level strategies [4]. Meta-analyses have demonstrated that multidimensional 34 medical interventions can effectively mitigate the risk of suicide [4]. The analysis of lan-35 guage and vocabulary of individuals attempting suicide allows for understanding the 36 thought process of the suicidal person, and consequently, offers a chance to anticipate 37 such an attempt [5]. ": It would be beneficial to mention the general trends of suicide or those identified by previous research in the introduction section. Suicide prevention strategies should be a topic addressed following this study.

2.  "Google Trends has been used to analyze public behavior during the COVID-19 pandemic [10]": I am not sure if this sentence is appropriate for the topic of this study.

3. Discussion section:

Have there been studies that analyzed Google Trends to confirm the prevalence of mental health issues in previous research?

Preceding studies related to Google Trends analysis of depression and severe stress could enhance the validity of this study.

4. Discussion section: 

What is the direction that the authors are aiming to convey through the results of this study?

Was there a correlation between the search volume of terms identified through Google Trends and the suicide attempt rate?

Based on the results of this study, are there any expected actions for future suicide prevention?

If previous research is lacking, look for other similar keywords (anxiety, depression, severe stress) in preceding studies.

Author Response

Dear Reviewer,

Thank you for your thoughtful and detailed feedback on our manuscript. We have addressed each of your comments as follows:

1. Introduction Section:

We have added information on general trends in suicide and highlighted the importance of prevention strategies.
Google Trends Usage:

2. Introduction Section:
The sentence "Google Trends has been used to analyze public behavior during the COVID-19 pandemic [10]" was indeed not fitting well with the rest of the content and has been appropriately adjusted.

3. Discussion Section:

Our work cites previous studies that have utilized Google Trends for analysis, including those highlighting its limited effectiveness in detecting suicides. It is important to understand that searching for terms online is only a small part of overall behavior, similar to non-verbal communication. Online behavior, such as the timing of activities and changes in these patterns, also needs attention. This study, as a separate component of the overall analysis of internet user behavior, is more fundamental in nature rather than applied. This has been further emphasized in the limitations section of the discussion.

4. Discussion Section:

We have clarified the novelty of our study and its findings in the discussion section. The value of this study lies in the large number of initial keywords and the fact that these words were not chosen empirically but were derived from an objective database. As per the editor's recommendation, the entire database will be provided as a supplement, allowing this publication and its database to be used for future research with adequate funding, which could analyze a larger dataset.

We believe these revisions have strengthened our manuscript and we appreciate your guidance.

Best regards,

Krzysztof Bartosz Klimiuk, 

Round 2

Reviewer 1 Report

Comments and Suggestions for Authors

I have no further comments.

Reviewer 3 Report

Comments and Suggestions for Authors

Your work on analyzing the correlation between online search behaviors and suicide rates is innovative and crucial. Utilizing data from the Polish Police Headquarters and Google Trends demonstrates a sophisticated understanding of real-time data in public health. This study enhances our understanding of suicide-related search trends and opens up new possibilities for developing predictive algorithms to identify at-risk groups. Your dedication to this important research has the potential to save lives and significantly improve mental health interventions.

Thank you.